# Making Historical Gyroscopes Alive—2D and 3D Preservations by Sensor Fusion and Open Data Access

**DOI:** 10.3390/s21030957

**Published:** 2021-02-01

**Authors:** Dieter Fritsch, Jörg F. Wagner, Beate Ceranski, Sven Simon, Maria Niklaus, Kun Zhan, Gasim Mammadov

**Affiliations:** 1Institute for Photogrammetry, University of Stuttgart, 70174 Stuttgart, Germany; 2Chair of Adaptive Structures in Aerospace Engineering, University of Stuttgart, 70569 Stuttgart, Germany; jfw@pas.uni-stuttgart.de (J.F.W.); kun.zhan@pas.uni-stuttgart.de (K.Z.); 3Institute of History, University of Stuttgart, 70174 Stuttgart, Germany; beate.ceranski@hi.uni-stuttgart.de (B.C.); maria.niklaus@pas.uni-stuttgart.de (M.N.); 4Institute of Parallel and Distributed Systems, University of Stuttgart, 70569 Stuttgart, Germany; Sven.Simon@ipvs.uni-stuttgart.de (S.S.); gasim.mammadov@ipvs.uni-stuttgart.de (G.M.)

**Keywords:** history of technology, computer vision, photogrammetry, endoscopy, computed tomography, convolutional neural networks, structure-from-motion, dense image matching, data fusion, sensor fusion, digital twin, navigation instruments, inertial sensors

## Abstract

The preservation of cultural heritage assets of all kind is an important task for modern civilizations. This also includes tools and instruments that have been used in the previous decades and centuries. Along with the industrial revolution 200 years ago, mechanical and electrical technologies emerged, together with optical instruments. In the meantime, it is not only museums who showcase these developments, but also companies, universities, and private institutions. Gyroscopes are fascinating instruments with a history dating back 200 years. When J.G.F. Bohnenberger presented his machine to his students in 1810 at the University of Tuebingen, Germany, nobody could have foreseen that this fascinating development would be used for complex orientation and positioning. At the University of Stuttgart, Germany, a collection of 160 exhibits is available and in transition towards their sustainable future. Here, the systems are digitized in 2D, 2.5D, and 3D and are made available for a worldwide community using open access platforms. The technologies being used are computed tomography, computer vision, endoscopy, and photogrammetry. We present a novel workflow for combining voxel representations and colored point clouds, to create digital twins of the physical objects with 0.1 mm precision. This has not yet been investigated and is therefore pioneering work. Advantages and disadvantages are discussed and suggested work for the near future is outlined in this new and challenging field of tech heritage digitization.

## 1. Introduction

The preservation of cultural heritage assets is an important task of modern civilizations. It provides identity, ensuring the understanding of the past, identification with traditions and customs, and allows for the accessing of existing, destroyed, or lost heritage objects.

Tangible assets of cultural heritage include technical instruments and artifacts—we call these tech heritage (TH). If these assets are historically researched and didactically processed, they allow for insights into developments and objects that have fundamentally shaped our civilization. Without professional processing, however, these assets remain silent; especially when they are technically complex and significantly encapsulated.

There are several methods and technologies for the 3D preservation of outdoors cultural heritage (CH) objects available and well-described in the literature—a most recent review is given by [1]. These are differentiated as active and passive sensing. Active sensors collect mostly range-based 3D data, by invasive direct measurements of mechanical systems or the operation of optical systems using triangulations, time-of-flight, or interferometry. Therefore, the active sensing technologies used so far in outdoors CH applications seem to be less suitable for the 3D preservation of TH objects.

This scenario completely changes when evaluating the methods and technologies of medical imaging. In medicine, a variety of methods for generating 3D volume data have been developed and are now widely used in medical applications to visualize the internal structures of human organisms, such as magnetic resonance imaging (MRI) (also called magnetic resonance tomography (MRT)), positron emission tomography (PET), and to an even greater extent X-ray-based computed tomography (CT). CT is nowadays used for a variety of sciences and applications beyond the medical field, such as material sciences, physics, biology, mechanical engineering, or technical applications, for the non-destructive determination of three-dimensional models of the internal structures of objects. For example, several studies in non-medical fields have been carried out by the authors of this paper and others using CT: the measurement of the properties of electrical structures [2], the application of dimensional metrology [3,4,5,6], and the contactless and non-destructive three-dimensional digitization and conservation of cultural assets in the context of digital heritage [7,8,9,10]. These non-medical applications are drivers for CT systems with ever higher spatial resolution, from the micrometer to the nanometer range, as well as for X-ray-based CT systems with ever higher photon energies, in order to irradiate materials that absorb much more strongly than tissues, such as metals and greater radiolucent material thicknesses. In summary, using medical technologies, such as CT, extends the existing methods of active sensing for collecting 3D data inside a TH object.

Passive sensing is image-based and uses natural light or enhanced lighting conditions. Based on the well-known methods and technologies for aligning and matching overlapping image blocks, as offered by geometric computer vision (CV) and photogrammetry, we also integrate endoscopy to collect overlapping images inside the TH objects (see Figure 1). The steps for all three fields are: first, these images are collected with calibrated DSLR cameras and endoscopes. Second, these images are aligned by structure-from-motion (SfM) or bundle block adjustment algorithms. Third, we accomplish dense image matching (DIM) using multi-view stereo (MVS) to get colored dense point clouds in 3D. Finally, the 3D colored point clouds are filtered and meshed to obtain “watertight” models.

The last twenty years have seen important milestones passed in the processing of stereo and multi-view stereo data, likewise in geometric CV and photogrammetry. The relation between photogrammetry and CV [11] has been considerably improved. It was proven that the projective equations of CV are identical with the collinearity equations of photogrammetry and therefore methods can be exchanged between them. Automatic image feature detection [12,13] provides automatic tie points for SfM (geometric CV) and bundle block adjustment (photogrammetry). In 2000 the 1st edition of a comprehensive collection of multiple view methods in CV was published [14]. Semi-global matching (SGM) was introduced by [15], leading to the enhanced quality of high resolution point clouds. An accurate, dense, and stereo reconstruction using SGM has been demonstrated by [16]. A refinement of SGM is tube-based SGM, which led to the development of SURE [17,18], a software for dense image matching (DIM) of airborne [19], close range [20], and most recently space-borne high resolution optical imagery [21]. Another refinement of the SGM algorithm is given by [22].

Preserving tech heritage in 3D through a combination of CT, CV, endoscopy, and photogrammetry is a new and fascinating field allowing for many options in historical research, education, and AR/VR applications. At the first stage, a 3D model of an instrument—inside and outside—by a meshed and watertight point cloud is offered, which can be shared by open access (OA) with a worldwide community. Thus, any intersections can be generated, using open source (OS) software, e.g., CloudCompare, MeshLab, the Point Cloud Library, and Pointools. A further processing stage is the decomposition of the model into Constructive Solid Geometry (CSG) features, e.g., vectoral elements, which represent all the parts in a Lego-like fashion. Other benefits of a CSG description is data compression and semantic labeling, i.e., putting attributes behind every geometric element. Up till now, CSG modeling with s satisfying output is accomplished only through very time-consuming manual work. In future, the methods of machine learning and deep learning may help to automate this decomposition, but this will not be reflected here. Therefore, with our research we are entering largely unexplored territory that holds the promise of many exciting developments in the years to come.

The structure is as follows: after the introductory Section 1 we describe the novelty of this work in combining 3D models of geometric computer vision with 3D voxels of computed tomography. In Section 2, for the first time we combine voxel clouds with point clouds to get a combined 3D model representing the interior and exterior object characteristics. Point clouds are normally fused and co-registered using the ICP algorithm, with no explicit information about their in-depth geometric qualities. We define a spatial similarity transformation embedded in a Gauss–Helmert model to estimate the variances of the unit weight and standard deviations of the combined data sets. Section 3 describes the University of Stuttgart’s gyroscopes collection, which was launched in the 1960s and contains about 160 objects. To sustain these TH assets, the Gyrolog project started 2017, with the mission to generate digital copies—we call these “digital twins”—using CT, geometric CV, and photogrammetry. In addition, the potential of endoscopy was to be explored. Section 4 deals with 2D photography and post-processing for easy object documentation and digital archiving. In Section 5 we outline the details of collecting 3D CT scans with denoising characteristics and 3D colored point clouds of CV/photogrammetry and endoscopy to finally create a 3D digital twin. The basic workflows and experimental results are presented. Section 6 contains the results of the 3D reconstructions of three different gyroscopes, including some historic research: (1) the Machine of Bohnenberger (1810), which is regarded as the very first gyroscope; (2) a directional gyroscope used for aircrafts in World War II (1940s); and (3) a gyroscope embedded in the inertial platform of the Lockheed F104G Starfighter (1960s). Moreover, two further examples demonstrate our capability to create 3D digital twins of the Stuttgart gyroscope collection. Thereafter their curation and sustainability in OA environments are outlined in Section 7. Finally, the conclusions and an outlook for future work complete this article, besides the references and list of abbreviations.

## 2. Merging 2.5D and 3D Data and Texturing—Making 3D Models Alive

In order to generate digital twins of tech heritage objects, such as the very first gyroscope invented by J.G.F. Bohnenberger in 1810 (see Figure 2), we decided on a combination of CT, endoscopy, and CV/photogrammetry. If we were to use CV/photogrammetry only, we would get the colored 3D hull of an object. Meshing the 3D points along the hull by 2D triangles and texturing it appropriately yields a 2.5D digital model. The definition of 2.5D is often used, when combing 3D coordinates with 2D topological elements, here triangles. As will be shown later, sensor fusion for this combination is only possible with endoscopy and CV/photogrammetry, as both generate optical image blocks to be processed by SfM and DIM in one processing step. This finally leads to consistent alignments and co-registered colored 3D point clouds.

The overall workflow of our novel approach of combining point clouds and voxel clouds is given by Figure 3. In summary, we are using two layers of data processing: the first layer delivers voxel clouds and point clouds and the second layer performs the data fusion of the CT, CV/photogrammetry, and endoscopy data.

The co-registration of CV/photogrammetry point clouds with 3D CT voxel clouds is only possible by data fusion using registration algorithms, such as the rigid body transformation or a spatial similarity transformation. Two intermediate steps are necessary: as we have CV/photogrammetry hull points we first have to find hull voxels. This is done by segmentation and ray tracing (see Figure 4a). The second step is a transformation of the processed voxel cloud to a point cloud (see Figure 4b). We have proven the two-step voxel-to-point cloud processing in [23] when coloring voxel clouds using photogrammetric hull textures. For the merging of two point clouds, the iterative closest point (ICP) algorithm [24] implemented in Open Source point cloud libraries, such as CloudCompare, MeshLab, PCL, and Open3D, is considered to be the classic approach. However, its disadvantage is that it is missing an in-depth quality measure for the registration. Thus, we define a seven parameter spatial similarity transformation embedded in an adjustment model, not only to co-register the 3D CT voxel clouds with the 3D point clouds of CV/photogrammetry, but also to achieve quality measures for the registration in the form of unit weight variances and standard deviations.

Starting with the seven parameter transformation with at least three control points, we get:***X*** = ***X_o_*** + *µ**Rx***(1)
where ***X*** is the (3 × 1)*_u_* vector of the target coordinates of *u* control points, ***X_o_*** is the (3 × 1)*_u_* vector of the three translation parameters (*X_o_*, *Y_o_*, *Z_o_*), *µ* is the scale, ***R*** is the (3 × 3)*_u_* rotation matrix depending on the unknown rotation angles *α*, *β*, *γ*, and ***x*** is the (3 × 1)*_u_* vector of the local *u* control point coordinates. This non-linear transformation is linearized considering only differential changes in the three translations, three rotations, and one scale, and therefore replaces Equation (1) by,
***dx*** = ***S******dt***(2)
where ***S*** is the (3 × 7)*_u_* similarity transformation matrix resulting from the linearization process of Equation (1), and,
***dt’*** = [*dx_o_*, *dy_o_*, *dz_o_*, *dα*, *dβ*, *dγ*, *dµ*](3)
representing the seven unknown registration parameters. In order to estimate the precision of the registration, a least-squares Gauss–Helmert model [25] must be solved, for *u* ≥ 3 and ***B*** = ***S***, leading to,
1st order: ***Av*** + ***Bx*** + ***w*** = **0**, and 2nd order: *D*(***v***) = *σ*^2^***P*^−1^**(4)

Solving Equation (4) with respect to ***v***, ***x*,** and the Lagrangian ***λ***, we use Gaussian error propagation for getting the desired dispersion matrices. With *D(**w**)* = *σ*^2^***AP*^−1^*A***’ the precision of the registration parameters is propagated to,
*D*(***x***) = *σ*^2^[***B’***(***AP*^−1^*A’***)^−1^***B***]^−1^(5)

*D*(***x***) contains the variances and covariances along its main diagonal and off-diagonals, which can be used to propagate any precision of the individual in-situ data collection method and finally assess the quality of the registration.

Let *V*(*O*) be the set of *m* voxels of a 3D CT scan and *P*(*O*) be the set of *n* points describing the object hull (3D) of CV/photogrammetry. First of all the ground sampling distance (GSD) or object sampling distance (OSD) of the CT scan should be similar to the CV/photogrammetry OSD. Then the CT hull voxels are identified and the whole voxel cloud is transformed to a point cloud, i.e., *V*(*O*) -> *VP*(*O*), representing a similar data structure of *P*(*O*) (see Figure 4). The next step is to choose *u* > 3 homologue points for the co-registration of *VP*(*O*) with *P*(*O*). The OSD of CV/photogrammetry for our applications is around 0.05–0.09 mm, and the CT scan OSD is about 0.06 mm. Thus the OSDs are similar but CT provides larger data volumes.

For the example in Figure 5, the CT scan of original resolution is about 86 GB and the CV/photogrammetry point cloud is about 425 MB. The down-sampling of CT by a factor of four reduces the data volume to 1.34 GB, and also the noise and artifacts. After an initial application of the ICP algorithm with a threshold of 0.05 mm using OS libraries, we chose *u* = 4 and *u* = 10 joint corners (control points) for the seven parameter transformations of the merged point clouds, finally obtaining improved registration results with the following quality measures for precision:

For *u* = 4 the estimated standard deviation of unit weight σ = 1.38, the estimated precision of the CT scans σ_CT_ = 0.10 mm, and the estimated precision of CV/Photogrammetry σ_CV_ = 0.08mm, and for *u* = 10 these results are σ = 1.07, σ_CT_ = 0.08 mm, and σ_CV_ = 0.06 mm. The more control points that are used, the better the precision can be estimated. These figures demonstrate that CV/photogrammetry 3D point clouds are more precise than down-sampled CT 3D volume data, and efforts are to be made to reduce noise and artifacts in CT scans (see Section 5.1) in order for them to arrive at the same level of precision. The results of our combined models and their corresponding precision values are given in Section 6. It is noted here that the data fusion of CT and CV/photogrammetry can provide digital twins with a precision of 0.1 mm.

## 3. The University of Stuttgart’s Gyroscopes Collection—The Gyrolog Project

Based on J.G.F. Bohnenberger’s 1810 invention of the gimbal mounted gyro, also called the Machine of Bohnenberger, and the work continued by J.B.L. Foucault to prove the Earth’s rotation, the theory of the gyroscope became a supreme discipline of physics during the 19th century [26,27]. The first technical applications followed at the beginning of the 20th century. Scientists like F. Klein, A. Sommerfeld, and M. Schuler at the University of Goettingen, and R. Grammel at the Technical University of Stuttgart, as well as entrepreneurs like H. Anschuetz-Kaempfe, Kiel, and W. v. Siemens, Berlin, made Germany a world leader in gyroscopes for flight and ship control, both scientifically and industrially. After World War II, K. Magnus, M. Schuler’s student and R. Grammel’s successor in Stuttgart, as well as the companies Anschuetz (Kiel), Bodenseewerk (Ueberlingen), C. Plath (Hamburg), Litef (Freiburg), and Teldix (Heidelberg) resumed this tradition [28].

Understanding the movement of the gimbal-mounted gyro and its derivatives is considered to be particularly challenging, both physically and mathematically [29]. To illustrate the teaching and research of K. Magnus, he and his assistants H. Sorg and J. Steinwand began in the 1960s to assemble a collection of gyroscopes at the Institute of Mechanics of the Faculty of Mechanical Engineering, University of Stuttgart. In 2005, the responsibility for this collection passed on to J.F. Wagner, of the Faculty of Aerospace Engineering and Geodesy at this University.

In addition to illustrating how gyroscopes and inertial navigation systems work, the collection also reflects the historical development of these instruments. It contains approximately 160 objects and includes most of the known types of gyro instruments for flight, land vehicle, and ship navigation (gyro compass, directional gyro, gyro horizon, P and I rate gyros, etc.) as well as various types of accelerometers. There are also complete inertial platforms. Components of the devices such as rotors, slip rings, and rotary encoders, as well as rotary tables for testing inertial sensors, are also available. Many exhibits were taken from decommissioned aircrafts and ships. Some of them have been cut open or partly dismantled, and some are still operational. They are mostly between 40 and 70 years old. This records the development of gyroscopic instruments, especially the work of H. Anschuetz-Kaempfe, Kiel, and E. Sperry, New York, up to the 1970s. The collection is unique in the higher education sector, at least in Germany, and is also complementary to other important collections of this type not only in Germany but worldwide. Some selected objects are shown in Figure 6.

In 2017, the German Federal Ministry of Education and Research (BMBF) awarded a grant for a project to create 3D digitization of this collection. The project is called Gyrolog (from the Greek γύρος, rotation, and λόγος, teaching). Its aim is to use digital methods (see Section 4, Section 5 and Section 6) to free the inconspicuous, yet highly complex objects, of today’s ubiquitous gyro technology from their black box in order to open this technology for research in the history of technology, technology didactics, museum education, etc., as well as for teaching in schools, universities, and institutions of further education. Digitization makes this technology understandable and in the truest sense of the word “virtually” and enables further research by the disciplines mentioned. The project was successfully completed at the end of 2020.

Furthermore, the virtual character of the digitized collection allows for the option of reuniting the collection with its historically formed subsidiaries at the Technical University of Munich, Germany, and the Johannes Kepler University of Linz, Austria. Additional instruments, such as an original copy of the Machine of Bohnenberger, which is described in more detail in Section 6.1, can also be added virtually.

## 4. Two-Dimensional Data Collections and Postprocessing

Meanwhile, the digitized collection of gyro instruments can easily be accessed by 2D pictures. This human–computer interaction interface provides first details of these highly complex and fascinating objects.

Accordingly, the intention behind the 2D digitization was to provide the user with as many views, as well as details, of the gyroscopic instruments as possible. However, compared to the newly created 3D objects, 2D pictures always provide limited object details depending on the field-of-view. The Gyrolog 2D setup is based upon photographic studio facilities, with a professional background and lighting. A light crème color was chosen as a background to contrast with the mostly dark, black, or metallic objects.

For the 2D data collection process a standardized procedure was created that included a fast response and digitization time as well as an elaborate way of handling the objects in order to minimize the impact of the data collection. First, the object is placed on the 2D set-up for the so-called characteristic view, which was developed together with a professional photographer. This characteristic view will guarantee a first impression with all the relevant information contained within the gyro instrument in one view, if possible, as seen in Figure 7a.

Then the instrument is rotated to display the front of the object. This process includes some challenges with regard to definition. Parts of the gyro collection are aircraft instruments that were built for and used in aircraft cockpits. Here the front view definition is quite easy. Other parts of the collection were rather more challenging but could also be defined together with our collection experts (see Figure 7b). Then the further data collection process was similarly executed as the object was carefully rotated in predefined ways: to display the left view, the rear view, and the right view. For the bottom and upward view, the instrument was cautiously turned. Furthermore, caging devices were used to minimize movements. These were individual fixtures that were partly 3D printed to fit to some of the objects within the collection (see also Figure 7a). In a last step, details such as type labels were digitized for a close-up view. With this data collection process the Gyrolog project ensured a detailed set of 2D pictures from every possible view of the individual gyro instruments.

This 2D dataset was also post-processed after a standardized sorting procedure. The basis for this was laid out during the digitization process. To distinguish between the different views, the 2D digitization photographer used different colored labels while digitizing the objects to support the post processing. The different labels were color-coded as well as labeled always in the same order: characteristic view, front view, and follow-up views. During the post processing the pictures were sorted into the different views for each instrument.

For every object, the best pictures are chosen and are available on the viewing platform Goobi. Goobi [30] is an open source, web-based software that is used by the University of Stuttgart’s library, a Gyrolog project cooperation partner, to ensure the sustainability of the digital gyroscopes collection (see Section 7).

## 5. Three-Dimensional Data Collections by Means of Computed Tomography, Computer Vision, and Endoscopy

With emerging technologies in the fields of data capture, data processing, and data visualization, three-dimensional object preservations have become state-of-the art practice for cultural heritage and also for tech heritage assets. One efficient and robust technology is photogrammetry, using horizontally and vertically overlapping photos to generate 3D reconstructions of surfaces and hulls. For a long time, photogrammetry served 3D mapping by means of bundle block adjustments and orthophotos. With the invention of DIM, photogrammetry underwent a renaissance and is equivalent and comparable to geometric CV. The photogrammetric bundle block adjustment is the pose estimation of computer vision, also called structure-from-motion (SfM). Large image blocks are automatically processed by SfM and DIM algorithms in one software package, delivering finally very dense colored point clouds which can be meshed for watertight models, also called virtual reality (VR) models or digital twins (DT).

Endoscopy is the use of imaging camera systems with huge enlargements but tiny fields-of-views (FoVs). Therefore, it is tricky to collect sufficiently overlapping image blocks through controlled camera movements along horizontal and vertical axes only. An endoscopic image block can be processed with the same workflow of geometric CV, moreover, it can directly be integrated into the CV image blocks which simplifies the point cloud registrations.

CT is a non-invasive imaging technology which directly provides 3D volumetric models or volume elements, in short voxels. As mentioned previously, the down-sampling of huge voxel files to reasonable data volumes is quite a challenge, which has to be overcome for the joint registration process.

The complete 3D contents of our gyroscopes’ DTs are obtained by an integration of CT voxel clouds with CV/photogrammetry and endoscopy point clouds. CT data and CV/photogrammetry point clouds are co-registered using the similarity transformation as described in Section 2. If endoscopic image blocks are simultaneously processed with the exterior image blocks, then they are automatically co-registered. If not, another similarity transformation ought to be accomplished.

### 5.1. Computed Tomography 3D Data Collection

In comparison to medical CT machines, industrial machines are more powerful and can easily penetrate through the different alloys of metals. In this work, an X-ray-based CT scanner with a resolution in the single digit micrometer range and with a photon energy up to 225 KeV is used, see Figure 8.

It is our concern to capture the interior of these historical gyroscopes in 3D without disassembling or destroying them. As is usual in CT systems, the 3D volume data is generated by rotating the sample placed on a turntable and taking several thousand X-ray images during the rotation; see Figure 8. From these projections the most widely used algorithm for CT reconstruction, the so-called filtered back projection (FBP) [3], is used. In selected cases, the maximum-likelihood expectation-maximization (MLEM) reconstruction algorithm is applied with a significantly higher computation time in order to generate 3D volume data of high quality in terms of noise of the objects [31].

There are several noise sources caused by artifacts in the 3D volume data. If the imaged gyroscopes are difficult to penetrate with X-rays due to their size and associated metal content, the projections are contaminated by massive Poisson noise.

To tackle this problem, we have applied several denoising techniques to the gyroscope samples. In order to show the results of the denoising, the X-ray images were taken with high current and acceleration voltage (Table 1) as a ground truth and later we simulated the strong Poisson noise over the scanned projections, where the approximate number of photons was about 500 per detector pixels.

In order to demonstrate this behavior, the target object KH-09-09 was chosen because the object has a massive, closed metal box structure and a large number of small features consisting of mechanical and electrical structures, which can be seen in the X-ray projection image in Figure 9.

Figure 9 shows that the noise is distributed over the entire projection and thus distorts the fine structure of the image. There are two methods that we have used to achieve noise-reduced projections that contribute to the final reconstructed volume. The first denoising method is based on a fully convolutional neural network (CNN), as proposed in [32]. The authors clearly show the benefit of the CNN for the presence of high Poisson noise, as is usually the case in CT, over the best denoising techniques available, with high number of photons and lower Poisson noise. The recently published CNN uses 20 connected layers, with 18 of them using Rectified Linear Unit (ReLU) nonlinearity and 2 of them using linear activation function. The architecture, presented in Figure 10, utilizes 64 kernels with size 3 × 3 to convolve on each layer. The network is trained on real X-ray projections with simulated Poisson noise. The network learns the unknown denoising function by training with known data sets. The result of the CNN is shown in Figure 11.

For the MLEM reconstruction we used our implementation combining the penalty which controls the total variation of the calculated volume at each iteration [31,33,34], including a regularization parameter *β*. The scanned specimen KH-09-09 was reconstructed with the regularization parameter *β* equal to 1 × 10^−2^ with a total number of 50 iterations. In the cases considered, the MLEM algorithm required about 50 iterations to reconstruct all features of the reconstructed volume. If the signal-to-noise ratio of the projections is low, the number of iterations should exceed 50. However, the MLEM algorithm has a computing time of more than 300 h on a powerful computer with 8 GPUs for the resolution of the detector used here for the projection images of 2300 × 3200 pixels. This is due to the large number of forward and backward projections. In order to reduce the computing time for the reconstruction significantly by a factor of 50 and still achieve high-quality volume data sets with noise reduction, denoised X-ray projections using the CNN architecture of Figure 10 were reconstructed using the FBP. For the CNN network training, a ground truth of the original high-dose X-ray images and reconstructed 3D volume data sets were generated using FBP. A denoised projection is shown in Figure 11. As can be seen from Figure 12, the results obtained from the MLEM reconstruction of the noise projections and the FBP of high-dose projections are similar, and in some regions MLEM produces better results, although it is very time consuming. Denoising images with the CNN algorithm helps to remove shot noise for high photon counts.

Thus, in comparison to the FBP of noisy projections, the FBP of denoised projections shows better overall results. The noise error for both methods can be calculated assuming the FBP of the original projections without noise as a ground truth. Thus, standard metrics such as the peak-to-signal-noise-ratio, root-mean-squared error (the standard deviation), and the signal-to-noise-ratio can be used.

For the object G200, we have generated the iso-surfaces for both, the denoising technique of the CNN architecture of Figure 10 and the iterative reconstruction algorithm MLEM. Compared to the first experiment, the Poisson noise will not be added to the projections. Thus, the reconstructed object will suffer from beam hardening, scattering artefacts, and shot noise. The scan parameters for the G200 specimen are given in Table 2.

Both the outer and inner structures of the G200 are complex. On the surface there are several cables and pins which cause an increase in streak artefacts as the walls of the object are thick and difficult to penetrate by X-rays, and are made of different metal alloys such as iron, copper, and aluminum. The main goal before scanning was to use a high level of energy and metal filters to harden the X-ray beam and prevent highly visible artefacts arising from the polychromatic nature of the source. Therefore, a copper filter with a thickness of 4.5 mm was used to reduce the beam hardening artefacts and exposure time of 1.4 s was used in order to receive more photons on the detector. The surface renderings [35] and iso-surfaces of the G200 object are depicted in Figure 13. As is obvious, the reconstructed volume based on the FBP has severe artefacts both on the outer and inner surface.

Figure 14 shows the difference in the iso-surfaces generated from the reconstructed volume from the denoised projections calculated by the CNN. The noise and metal artefacts are significantly reduced in comparison to Figure 13, where the denoising technique was not applied.

In comparison to Figure 13, Figure 15a has been reconstructed by down-sampling the projections by a factor of four. The quality of the iso-surface is further significantly improved due to the increased signal-to-noise-ratio produced by the down-sampling process in each dimension. This factor of four results in an increase in the number of photons of 64 per voxel in the 3D volume, which enhances the signal-to-noise-ratio in dB by a factor of eight, which is also a reason for the overall improvement in quality of the iso-surface view in Figure 15b compared to Figure 14. The reduction of the spatial resolution by down-sampling is no limitation in the context of the CT images considered here, since voxel sizes after down-sampling by a factor of four are still in an acceptable range below 250 µm, which allows a sufficient resolution for the model computed by the data fusion of CT and CV/photogrammetry data. This resolution is no limitation for the use of the data to visualize the objects including their internal structures in the context of digital heritage. The denoised projections based on the FBP reconstruction of Figure 15b show the best overall results of all the considered cases. Obviously, based on the iso-surface, we can accurately differentiate the structure of the Regions-of-Interest (RoI) indicated by the square and circle. Both are parts of the electronic components and the rotor shape, respectively.

Based on the methods described above, 3D models of the gyroscopes in the collection were generated. As an example, parts of the Machine of Bohnenberger (see Section 6.1) are reconstructed and shown in Figure 16. The structure of the inner rod can be clearly seen in a cross-sectional image of the 3D volume data set in Figure 16b. In addition, a further segmentation with a predefined material threshold results in an iso-image of the entire surface of the gyroscopic gimbal (see Figure 16c).

### 5.2. Geometric Computer Vision and Photogrammetric 3D Data Collection

The principle of three-dimensional (3D) point reconstruction from imagery is called triangulation (see Figure 17). An object is captured in at least two different photos and the corresponding image point coordinates of p_1_ and p_2_ are measured. With additional camera orientation information for O_1_ and O_2_ (image poses) the 3D corresponding point P can be calculated by the forward intersection.

The 2D–3D correspondences can be either expressed by the collinearity equations in photogrammetry:(6)$$x-x_{0}=-f\frac{r_{11}\left(X-X_{0}\right)+r_{21}\left(Y-Y_{0}\right)+r_{31}\left(Z-Z_{0}\right)}{r_{13}\left(X-X_{0}\right)+r_{23}\left(Y-Y_{0}\right)+r_{33}\left(Z-Z_{0}\right)}\;y-y_{0}=-f\frac{r_{12}\left(X-X_{0}\right)+r_{22}\left(Y-Y_{0}\right)+r_{32}\left(Z-Z_{0}\right)}{r_{13}\left(X-X_{0}\right)+r_{23}\left(Y-Y_{0}\right)+r_{33}\left(Z-Z_{0}\right)}$$
or as the projective equation in computer vision:(7)$$\left[\begin{array}{c} x\\ y\\ 1 \end{array}\right]=\left[\begin{array}{ccc} f & 0 & x_{0}\\ 0 & f & y_{0}\\ 0 & 0 & 1 \end{array}\right]R^{T}\left[\begin{array}{cccc} 1 & 0 & 0 & -X_{0}\\ 0 & 1 & 0 & -Y_{0}\\ 0 & 0 & 1 & -Z_{0} \end{array}\right]\left[\begin{array}{c} X\\ Y\\ Z\\ 1 \end{array}\right]$$

The notation of CV can be abbreviated as
(8)$$x=K\left[R|t\right]X$$
where K is the calibrated camera matrix.

In practice, the CV/photogrammetry workflow is split into several steps [13,14,16,18,20], as shown in Figure 18a. First of all, we work with calibrated camera systems, no matter if we use a DSLR camera, an “off the shelf” camera, or cell phone cameras. Camera calibration has been an issue for photogrammetry for about 100 year, with very precise calibrations for close range applications proposed in the 1960s. In this project we investigated the performance of camera calibration for four camera systems: the Gyrolog project camera Sony α7R II, a Leica Q, an Apple iPhone 7Plus, and a Samsung Note 8 [36]. When extending the ideal lens characteristics with distortion, the collinearity Equation (6) are reformulated as:(9)$$x-x_{0}=-f\frac{r_{11}\left(X-X_{0}\right)+r_{21}\left(Y-Y_{0}\right)+r_{31}\left(Z-Z_{0}\right)}{r_{13}\left(X-X_{0}\right)+r_{23}\left(Y-Y_{0}\right)+r_{33}\left(Z-Z_{0}\right)}+\Delta x\;y-y_{0}=-f\frac{r_{12}\left(X-X_{0}\right)+r_{22}\left(Y-Y_{0}\right)+r_{32}\left(Z-Z_{0}\right)}{r_{13}\left(X-X_{0}\right)+r_{23}\left(Y-Y_{0}\right)+r_{33}\left(Z-Z_{0}\right)}+\Delta y$$

Here Δx and Δy are the correction terms for the image coordinates, $r_{\mathrm{ij}}$ are the components of the rotation matrix **R**. With regards to camera distortion, various models are based on either the mathematical principle, the physical principle, or the mixed principle. Among many, the classical Brown model and its variants are most widely used [37]. It classifies the distortion into radial distortion ∆_r_ and tangential distortion ∆_t_.
(10)$$\Delta =\Delta_{r}+\Delta_{t}$$

In the Brown model, the radial distortion is modeled by the three parameters *K*_1_, *K*_2_, and *K*_3_, and *P*_1_ and *P*_2_ are tangential distortion parameters. Furthermore, $r=\sqrt{u^{2}+v^{2}}$:(11)$$\Delta u=u\left(1+K_{1}r^{2}+K_{2}r^{4}+K_{3}r^{6}\right)+\left(2P_{1}\mathrm{uv}+P_{2}\left(r^{2}+2u^{2}\right)\right)$$
(12)$$\Delta v=v\left(1+K_{1}r^{2}+K_{2}r^{4}+K_{3}r^{6}\right)+\left(2P_{2}\mathrm{uv}+P_{1}\left(r^{2}+2v^{2}\right)\right)$$

Various methods based on the geometrical relationship are put forward, with regards to calibration scenes, calibration models, and estimation processes. For this work, experiments are mainly dependent on the Matlab^®^ Calibration Toolbox, which uses a planar chessboard.

The CV/photogrammetry data acquisition process is on display in Figure 18. First, we start with camera calibration. An example of the calibrated focal length (pixel) in x direction for the DSLR Sony α7R II is shown in Figure 19. For the four calibrated cameras, the Gaussian fitting standard deviations of the calibrated focal length are shown in Figure 20.

After the camera calibration, the data acquisition of the gyroscope objects begins. The object to be digitized is placed on a turntable under an appropriate lighting configuration, and the camera is fixed in a suitable position to take pictures, while the turntable is rotated at a well-defined speed (see Figure 18b). With 500 to 800 images from all views in horizontal and vertical modes, and after estimating the camera pose using SfM, a dense point cloud can be calculated by DIM for further 3D modeling processes or VR/AR (virtual and augmented reality) animation. When the calculated 3D model is not complete, most probably due to the lack of information from invisible perspectives, additional images need to be taken and the corresponding point clouds will be integrated with the previous ones to complete the model. This process is called point cloud registration and is part of the process shown in Figure 18a. A first successful test coloring 3D CT scan with photo textures is given by [37].

### 5.3. Endoscopy in 3D

An endoscope is an illuminated optical, typically slender, and tubular instrument. The lens projects a real-life scene image onto the first focal plane, which is then transmitted by the reversal system to the final focal plane. At the eyepiece, the image is projected onto the file by a camera lens. An example of an endoscope can be seen in Figure 21. Among all steps, the proper acquisition of data delivers the biggest difference in comparison to normal camera applications. In addition to the visual effects of an endoscope, more factors should be taken into account, such as accuracy, image quality, appropriate image blocks, and many more.

In traditional 3D reconstructions within CV and photogrammetry, the overlap between neighboring images should be over 80%, due to imaging distance and resolution. A DSLR or off the shelf camera fulfills this requirement without much effort, however for an endoscope, the situation is quite different (see Figure 22).

Suppose the imaging distance is d, the coverage of the object space can be calculated via
*L* = 2*d* × *tan*(*α*/2)(13)

In our case, the viewing angle is 75° and the imaging distance ranges from 5–15 mm. Therefore *l* can be determined from 8–23 mm. If we need an 80% overlap between the neighboring images, the movement of the tip should be 1.6–4.6 mm, which is extremely difficult to accomplish in practice while operating the endoscope to collect image blocks.

Due to the challenging imaging characteristics stated above, a suitable endoscope type should be chosen to ensure invasive possibility and sufficient image quality. In addition, using images only as an input for pose estimation requires suitable image configurations, especially for the overlap between neighboring images. The endoscope holder can be either mechanical, which gives less automation, or a robotized device (which is normally more expensive). Since highly precise movement control is hard to achieve in practice, a streaming video can be used in the place of taking still images. In practice, though video frames deliver lower resolution than endoscopic still images, highly overlapped extracted frames can make the task of image alignment much easier. Even with streaming video as input for 3D reconstruction, there are still several necessary precautions that should be undertaken.

Therefore, instead of using free hand movements, a more stable solution is proposed using a self-designed gimbal. This fixture has four degrees of freedom and the whole system consists of several parts: for horizontal movement, a cross slide is used as the base, and another single slide is attached via a 3D-printed 90 degree adapter with the cross slide. In addition, a ball head is fixed for the rotation movement. With the assistance of this system, it is possible to use screws to calmly move the endoscope precisely to create enough overlap between the images. The endoscope is thus able to move forward and backward, left and right, up and down, and also laterally.

The movement of the gimbal-assisted endoscope is shown in Figure 23. Here a total of about 200 overlapping endoscopic images have been processed to deliver a colored point cloud (the software used for SfM and DIM is RealityCapture).

Finally a comparison of the 3D meshes generated by endoscopic and photogrammetric image blocks can be accomplished (Figure 23b,c). Due to the enlargements of the endoscopic camera systems more details can be resolved than with regular DSLR or off-the shelf cameras, which might be important for the historical research of a gyro system. Working with endoscopic image blocks is a challenge. With the self-designed mechanical gimbal the generation of densely colored point clouds has been proven to be feasible. Further experiments will follow that explore the full potential of a multi-view stereo of large endoscopic image blocks.

## 6. Results for Digital Twins of the Gyroscope Collection

### 6.1. The Machine of Bohnenberger—The Very First Gyroscope

In 1810, the astronomer, mathematician, and physicist Johann Gottlieb Friedrich Bohnenberger (1765–1831) invented the gyro with cardanic (or gimballed) suspension at the University of Tuebingen in south-west Germany [27]. This instrument served initially as a teaching tool during his lectures in astronomy. Bohnenberger used it for demonstrating the orientation of the Earth’s rotation axis during its yearly journey around the sun, as well as for demonstrating the precession motion of this rotation axis (the latter is a periodic movement with a duration of about 25,800 years, while the axis describes the surface of a double cone. This effect leads to a slow circular motion of all the fixed stars in the night skies over that long period). Bohnenberger described systematically the design and the use of this instrument for the first time [38].

Based on the details of Bohnenberger’s paper, it was possible to discover two original copies at a school in Tuebingen in 2004 and during an internet auction in 2010—the only originals still in existence known to the authors. These findings gave rise to historic research about the early dissemination of the instrument and its contribution to modern gyro technology: probably in 1812, Bohnenberger had already sent two copies of the instrument to P.S. Laplace and the Ecole Polytechnique in Paris, respectively. In subsequent years this apparatus was introduced in the physical collections of many French schools. Therefore, L. Foucault knew this device very well. During his investigations on a simple experimental proof for the rotational motion of the Earth he improved Bohnenberger’s invention and called it gyroscope. The reason for Foucault’s occupation with this device was that he could not get sufficient approval of his well-known pendulum, as this instrument can sense the vertical component of the Earth’s rotation rate [39] only. A gimballed gyro, however, is at least theoretically applicable for measuring the full rate. Although technical imperfections prevented Foucault from being successful with this approach, his work opened up the development of important navigation instruments, in particular, the gyro compass, artificial horizon, and directional gyro (leading also to contemporary inertial and integrated navigation systems) [26,27,28]. Furthermore, it is this success of the gyro with cardanic suspension which stimulated decisively the development of Laser gyros and fiber-optical gyros, as well as micro electro-mechanical (MEMS) gyroscopes, as a mass product of today.

The Machine of Bohnenberger is virtualized in 3D by a combination of 3D CT scans and 3D CV colored point clouds, using the similarity transformation of Section 2. In total, about 500 photos were taken with the Sony α7R II DSLR camera and processed by AgiSoft’s PhotoScan, now Agisoft Metashape, and Capturing Reality’s RealityCapture software packages. A comparison of both the photo alignment and DIM softwares shows a clear advantage for RealityCapture, as this software is faster and resolves for DIM in critical lighting conditions. The 3D digital twin is on display in Figure 24.

This digital twin of the Machine of Bohnenberger can be used for augmented reality and virtual reality animations, as well as for reverse engineering and, perhaps most interesting, for 3D printing. In addition, it can be easily decomposed into its basic components by 3D constructive solid geometry (CSG) modeling, for which every CSG feature can be semantically enriched to provide an overview of the functions and materials.

### 6.2. Directional Gyro LKu4

A rather different example for bringing a mechanical gyroscope alive is the directional gyro LKu4 manufactured by Siemens. This instrument was built during the 1930s and 1940s in Germany, thus being the gyro instrument used extensively by the German armed forces during World War II. Furthermore, it makes this directional gyro an early example of mass-produced gyro instruments within Germany’s wartime production. In the Stuttgart collection, one finds several of these objects. There is an unaltered one (KK02-09), one that has been cut open (KK12-09) to show students the inside and the working principle of the instrument, as well as on that has been altered (KK09-09) to show the object in running conditions. In addition, there is another model available that is being used for a research project of a diploma thesis [40] (KK34-20), and several other objects (KK17-09, KK22-09) containing parts of this kind of directional gyro. These different usages of the gyro objects demonstrate a distinguishing feature of university collections in comparison to museum collections—the use of the objects in research and teaching.

To follow the path of these objects, one has to start any historical research with the object or its digital twin, as there is nearly no written information preserved which gives testimony regarding their origin or their use before they entered the Stuttgart collection. Due to oral history interviews conducted as part of the Gyrolog project we know that some Siemens objects might have come from surplus war material directly from the Siemens company, after WWII. The university institute, in which the collection was established during the 1960s, was located right next to the building of the Siemens Stuttgart branch [41].

The LKu4 was part of a range of products produced by Siemens during WWII. This model and all its different variations were widely used within the German air force, for example in some versions were found in the Heinkel He111 and Junkers Ju88 [42] (see Figure 25a). It was one of the first electrically powered directional gyros (in comparison, for example, to the pneumatical directional gyro mainly used by the American air force) and could also be used within a yaw-axis control mechanism. Thus, this object is a good example of different technological developments in different countries. Siemens pointed out the advantages of electrically driven gyros in its advertisement (see Figure 25b), e.g., that the risk of freezing in higher altitudes [43] is reduced. The Stuttgart object is displayed in Figure 25c.

As the LKu4 has transparent and mirroring material at the front, the CV/photogrammetry 3D reconstruction process is a bit different from the conventional workflow. Here we used not only ordinary photos taken with the turntable, but also sprayed front-view images. After the data acquisition, all image poses are estimated together in the same coordinate system. In the mesh reconstruction phase, ordinary front-view images are deactivated while only sprayed ones are used for the DIM. The result is given in Figure 26b.

Looking at the Gyrolog objects and comparing them with later models, one can notice that they are all from an earlier manufacturing line (presumably before 1944) as they all still have a bank indicator and internal lighting [44] (see Figure 25c). One can partly trace these timelines by looking at the manufacturer as indicated by the type labels of the objects mentioned above, seen in Table 3. One realizes that these four objects were produced by two different manufacturers actually of the same company, but in different departments. In 1936 the “Abteilung für Luftfahrtgeräte” was established within the Siemens Apparate und Maschinen GmbH (SAM). Between 1936 and 1938, plans were developed to establish a central department within Siemens only for aircraft equipment: the Luftfahrtgerätewerk.

“Hakenfelde” as an addition to the name was added to indicate the location, a part of Berlin-Spandau, where the department was situated. The LGW (Luftfahrtgerätewerk Hakenfelde GmbH) was built in Hakenfelde between 1939 and 1941 and officially established as a sub-company of Siemens on 1 October 1940. This meant that all activities under the name SAM were ceased and reestablished under the name LGW [46,47,48]. This means that only KK09-09 was probably manufactured before late 1940, and the other three were manufactured presumably between late 1940 and 1944.

### 6.3. Gyro G200 of Inertial Platform LN3

Another example is the LN3-G200 gyro in the collection, with the inventory number LK05/01-17 (see Figure 27). It was part of an inertial platform, the so-called LN3, manufactured by Litton Technische Werke, Freiburg (Litef). The German part of the American company was founded for the purpose of producing the LN3 platform for the Lockheed F104G Starfighter, purchased by the German government during the Cold War Period.

This object’s story reflects the exemplarily lived history of the Gyrolog collection and also traces the links between manufacturer, the state, and universities. Thus the object is embedded in this so-called triple helix structure [49,50]. In 1968 the G200 gyro was being transported in a special transportation box (LK05/02-17) (see Figure 27b) to the Institute of Mechanics at the University of Stuttgart and was supposed to reach Dr. Ing. Walter Schmid, an associate of the institute. However, it was delivered to Professor Helmut Sorg on 27 September 1968. Additionally, part of this delivery included not only these two objects but a folder with information about this particular gyroscope [51]. It contained technical test documents (such as balance tests, gyro torque tests, etc., for different tests benches), which were carried out by Litef during January 1968 until September 1968. Obviously, they were neither carried out for W. Schmid nor H. Sorg nor the University of Stuttgart, but for the Federal Office of Armed Forces Technology and Purchases (Bundesamt für Wehrtechnik und Beschaffung, BWB), an institution of the Federal Ministry of Defense. The object shows this, as it is indicated in the sources that the object has a BWB number (BWB 1039), a military supply number (6615-00-754-4920), and the information that it came from a military depot (BEL—Bundeseigenes Lager) as well as the indication on the delivery receipt that the G200 gyro was part of an LN3-2A platform in a F104G. H. Sorg and W. Schmid were both members of the initial institute that started this collection under the direction of Professor Magnus. In 1968, Professor Sorg was the interim head of the institute during a period when Professor Magnus left to the Technical University of Munich. W. Schmid finished his dissertation [52] as well as the report for the German Science Foundation (DFG) on the same topic in December 1967.

Due to these objects and archival sources, one has a starting point for any historical research into the connections between the research, manufacturing, and use of gyro instruments in the 1960s in the German Federal Republic. The G200 gyro can be placed in the wider context of a complex structure between these three branches within its development: a university (University of Stuttgart), an industrial manufacturer (Litef), as well as the German Federal Republic, here in form of its military branch (BWB—Ministry of Defense).

The 3D digitization of this object and its historical contextualization was only the first step in bringing this gyroscope object back to life. The next step was the 3D modeling to animate its functional principle in an AR smartphone application. In a master’s thesis within the Digital Humanities Department at the University of Stuttgart the object is rebuilt digitally based on the 3D data generated by the Gyrolog project. As is well-known, the G200 gyro is situated in the complex story of the Starfighter-Affaire, an interesting and a media-savvy political drama within Germany’s Cold War history [53]. This, as well as its functional principles as a black box navigation tool, makes these instruments very interesting for the history of science and technology museums. The Gyrolog project together with the master’s thesis have been cooperating with the Deutsche Museum, Munich, that will incorporate this final 3D application into its new tour after the reopening of the aircraft exhibition, thus bringing this gyroscope digitally back to life for many visitors from all over the world.

The 3D digital twin of the G200 Gyro is on display in Figure 28, where Figure 28a is the 3D CT scan, Figure 28b the 2.5D CV/photogrammetry reconstruction, and Figure 28c a split view into the CT/CV integrated 3D model.

### 6.4. Two more Examples of Gyrolog 3D Digital Twins

In order to demonstrate our workflows and capabilities we were able to produce 3D digital twins of all sizes of the TH assets in a cube box range with up to 0.3 m edge length. Our gyroscopes have a maximum size of 0.1^3^ m^3^ with very complex structures, such as metal sheets, wires, electrical drives, and servos, etc. The view into the TH object can be accomplished without any difficulties using OS libraries, e.g., CloudCompare, MeshLab, PCL, etc.

In Figure 29, the pneumatically driven gyro, manufactured by Ternstedt Manufacturing Div., GM Corp., Detroit, USA (see also Figure 6a) and the electrical direction gyro, manufactured by Siemens LGW, Berlin, (see also Figure 6b) are on display. Looking at all of the three views—the internal view obtained by CT scans, the CV/photogrammetry external view, and the split view of the integrated model—the complexity of the gyro instruments is underlined.

A summary of the precision values for the Gyrolog data fusion of CT voxel clouds and CV/photogrammetry point clouds is given by Table 4. When using at least ten control points we come close to the ideal value of σ_0_. As stated below, all 3D digital twins could be reconstructed within a precision in the range of 0.1 mm, which is quite sufficient for our applications.

An interpretation of Table 4 is as follows: using four control points only for the data fusion process of CV/photogrammetry point clouds with CT voxel clouds, the standard deviation of the unit weight deviates up to 38% from the nominal value (1.0). Thus ten control points improve the data fusion process and the precision values considerably: σ0 comes close to the nominal value and the standard deviations of the individual data collections are about 0.1 mm. The quadruple down-sampling of the CT resolution (GSD = 0.22 mm) for the G200 data collection has not had a considerable impact on its precision σCT = 0.08 mm. The file sizes are given in **.obj* data format and are reconsidered by Table 5.

## 7. Curation of Gyrolog and Open Access

Stable long-term accessibility of both the digitized objects as well as their pertinent metadata are decisive prerequisites for the sustained usefulness of all digitization efforts—otherwise invisible objects would merely be replaced by hidden or, even worse, lost digital data. In the Gyrolog project, sustainable data curation rests on four pillars addressing the technical and the formal aspects, as well as the administrative aspects, of longtime accessibility.

First of all, the Gyrolog project maintains a strategic partnership with the university library to grant stable availability of the digitized objects. This institution has already accumulated considerable expertise with the 2D digitization of books, architectural drawings, maps, and the like. The library presents its digitized resources via the viewing platform Goobi [30]. Goobi is an OS web-based software providing both the front-end viewer as well as the back-end digitization management system. In order to present the Gyrolog data, the viewer has recently been expanded to include 3D representations (see Figure 30).

It uses *glTF* data format for 3D graphics output, whereas **.obj* files are used in Goobi’s internal workflow. The graphics language Transmission Format (*glTF*) was developed by the Khronos Group 3D Formats Working Group and minimizes both the size of the 3D objects and the runtime needed to unpack the objects. A difficulty that occurred during the implementation points to a characteristic dilemma of digitization: digitalized objects become unmanageable if they are too detailed because data files will become too big for comfortable consultation. 

The huge **.obj* files are significantly compressed when being turned into **.gltf* files, but in most cases additional compression via the open source compression library Draco has proven indispensable to ensure a smart user experience. Table 5 displays the file sizes of the specified Gyrolog objects presented in this paper. Thus, the *gltf*/Draco format is recommended for the 3D data sharing of our Gyrolog 3D digital twins.

Gyrolog is the pioneering project for Goobi’s enhanced functionality and it is expected that several other university libraries will follow. Contrary to museums, university collections in Germany usually have no permanent professional IT personnel of their own and often are an institutional subunit within the local university library.

In contrast, Goobi’s curated data workflow allows for both 2D and 3D representations including the pertinent metadata, and Gyrolog’s massive amount of research data is hosted by DaRUS, the University of Stuttgart’s data repository for long-time data storage and accessibility. Via DaRUS, which forms the second pillar for Gyrolog’s longevity, all available 2D data as well as the CT and CV/photogrammetry data will remain accessible for at least ten years. This would enable researchers in the future to reprocess point clouds, e.g., with novel algorithms. The file system strictly refers to the objects’ inventory numbers and allows for the unambiguous attribution of every data set to the original object. This is all the more important as the data comes from many different sources and is in different formats. The inventory thus forms the data hub of Gyrolog’s metadata stock, the third pillar of long-term searchability. Apart from the images’ pertinent technical metadata, it contains semantic metadata providing information on the respective object’s manufacturer, its life cycle, and many more aspects. Curating the data comprehends the development of controlled vocabulary—for example the common labeling as “gyro” obviously is no help for the differentiation of the objects in the collection and a more finely tuned classification system had to be developed. This was done in cooperation with the Deutsches Museum at Munich, another facility holding a major collection of gyros in Germany. The unambiguous identification of manufacturers, corporations, and scientists/inventors is achieved by reference to the Integrated Authority File (in German, Gemeinsame Normdatei, GND) [54]. Conceptual conformity with the CIDOC Conceptual Reference Model and metadata format interoperability with major web portals such as Europeana (via the Deutsche Digitale Bibliothek to which Goobi exports data routinely) allow for maximum accessibility to Gyrolog’s digital data.

Accessibility, finally, refers not only to formal issues such as controlled vocabulary and technical aspects such as longtime data storage, but also to legality. Here, open access (OA) is the fourth pillar of Gyrolog’s sustainable data curation. All Gyrolog data and metadata are shared under the creative commons license CC-BY-SA. Open access was not only a formal requirement for the grant but is also requisite for the intended broad use of the digital twins in teaching and research, for example in the history of technology or for visualization purposes in courses on mechanics. Moreover, crowdsourcing has become a substantial human resources supply in developing cultural heritage [55]. There are so many knowledgeable gatherers and enthusiasts of gyro instruments whose expertise might prove valuable in filling the gaps in our knowledge of our objects and of gyro history more generally. To get to know these experts and to encourage them to contribute to Gyrolog’s semantic metadata is an exciting task that is only just beginning as we approach the termination of the proper digitization process.

## 8. Conclusions and Outlook

This paper has demonstrated that the 2D, 2.5D, and 3D digitization of tech heritage objects is a challenge that can be mastered through the combining of different technologies. For our application, in order to sustain the gyroscope collection of the University of Stuttgart, we used computed tomography, geometric computer vision, endoscopy, and photogrammetry. First, all of the objects were 2D photographed and labeled, for archival purposes. The real challenge lay in the generation of the 3D digital twins. As is well-known, CT delivers 3D voxels of superb resolution, much higher than the ground sampling distances (GSDs) of CV and photogrammetry. Thus, a decision was made to resolve CT models with lower and similar GSDs as used in CV/photogrammetry, but with denoising characteristics. It was also proven that endoscopic image blocks can be aligned in one structure-from-motion processing step, to estimate the poses of all images. This is a big advantage, although the image block data collection using endoscopes is difficult to maintain. Afterwards, pose estimation disparities for pixel-wise dense image matching are calculated and point clouds are derived. The fusion of CT 3D voxel clouds and CV/photogrammetry 3D colored point clouds is accomplished by a spatial similarity transformation embedded in the Gauss–Helmert model of statistical inference. The advantage of this fusion compared with classical ICP solutions is the quality assessment. Here the standard deviations of the fused 3D models are clear indicators for the goodness-of-fit of the CV-CT/photogrammetry data fusion process. The final model is three-dimensional and reconstructed with a precision of about 0.1 mm.

Finally the 2.5D and 3D digital twins are made open access in the *gltf* format, using the Goobi and the DaRUS platforms. Through the Library of the University of Stuttgart the maintenance and sustainability of the digital twins is secured for a period of ten years. This means researchers from all over the world can download the 2D photos, 2.5D and 3D digital twins, and the object semantics for their own research.

Our research is a first step into the 3D digitization of tech heritage and we are proud of the results achieved. The next steps are the Lego-wise decompositions of the complex gyroscopes using constructive solid geometry modeling. So far, we have decomposed simple structures such as the Machine of Bohnenberger and other surveying instruments using Maya, Blender, and Autodesk 3ds MAX in time-consuming manual work. To apply machine learning and deep learning methods to get similar quality by automatic decompositions compared with manual work provides an intriguing challenge, however this will take most probably another decade.

## Figures and Tables

**Figure 1 sensors-21-00957-f001:**
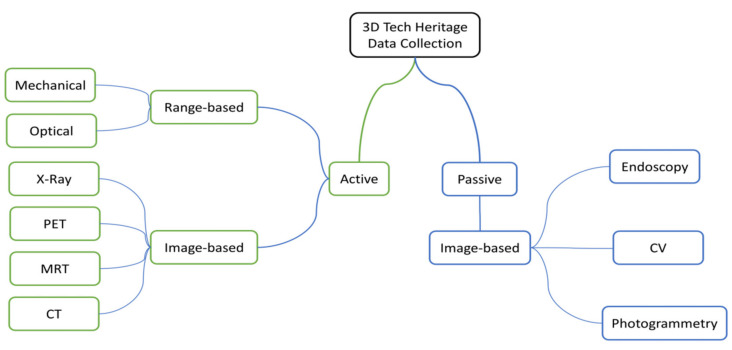
Taxonomy for 3D tech heritage data collections.

**Figure 2 sensors-21-00957-f002:**
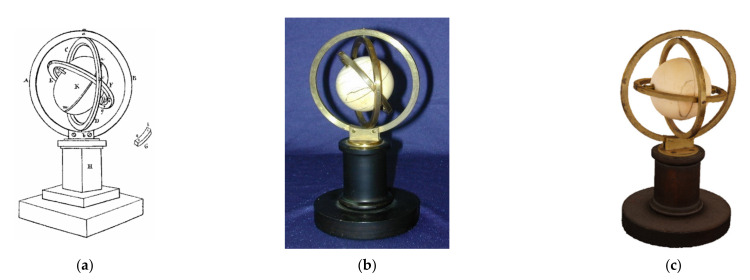
The very first gyroscope of 1810—the Machine of Bohnenberger, Tuebingen, Germany. (**a**) Original drawing; (**b**) Photo; and (**c**) 2.5D digital twin by CV/photogrammetry.

**Figure 3 sensors-21-00957-f003:**
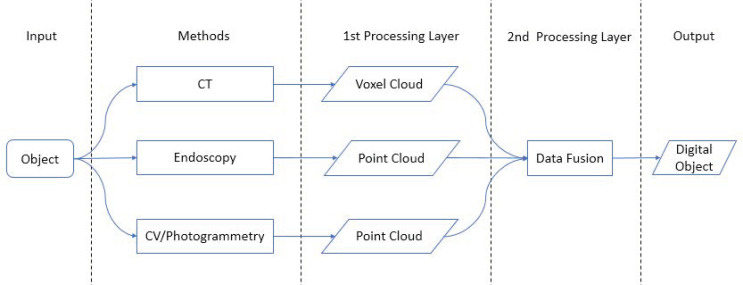
Processing pipeline of the Gyrolog project.

**Figure 4 sensors-21-00957-f004:**
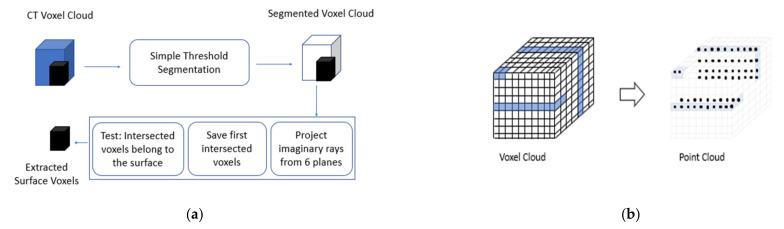
Voxel-to-point cloud transformation: (**a**) Step 1: find surface voxels; (**b**) Step 2: voxel-to-point cloud transformation.

**Figure 5 sensors-21-00957-f005:**
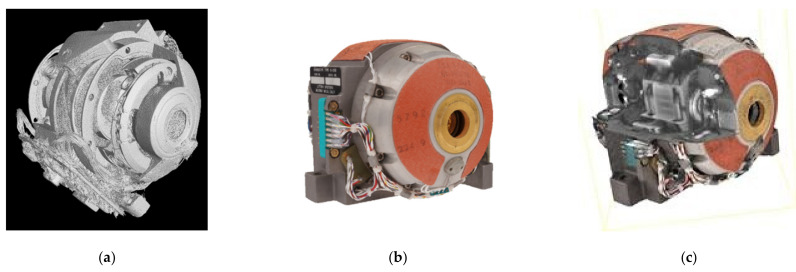
Example of the co-registration of CT Scans with CV/photogrammetry point clouds—the 3D digital twin of the Gyro200 (see Section 6.3): (**a**) iso-surface of a denoised filtered back projection (FBP) CT scan; (**b**) the convex hull generated by CV/photogrammetry; and (**c**) a cross-section of the integrated CT and CV/photogrammetry 3D reconstruction.

**Figure 6 sensors-21-00957-f006:**
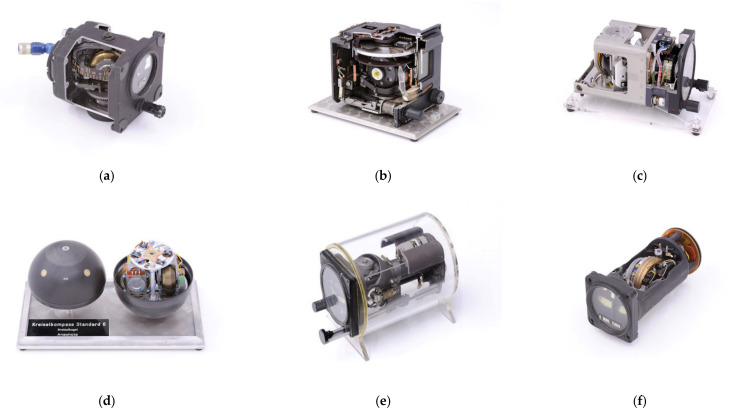
Examples of the University of Stuttgart’s gyroscope collection (Photos: B. Miklautsch, Photography Lab, University of Stuttgart, 2010): (**a**) pneumatically driven direction gyro, Ternstedt Manufacturing Div, GM Corp, Detroit, USA; (**b**) electrical direction gyro by Siemens-LGW, Berlin, Germany; (**c**) direction gyro S.F.I.M by BEZU, France; (**d**) gyro compass of Anschuetz, Germany; (**e**) artificial Horizon, manufacturer not known; and (**f**) electrical turning pointer by Apparatebau Gauting, Germany.

**Figure 7 sensors-21-00957-f007:**
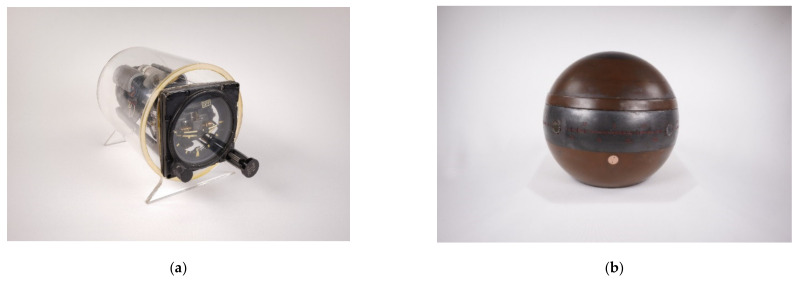
(**a**) Characteristic view of the artificial horizon with a caging device (inventory number KH22-10, Gyrolog, CC-BY-SA). (**b**) Difficult definition of a front view of a ship gyro compass manufactured by Anschuetz (inventory number GO05/01-10, Gyrolog, CC-BY-SA).

**Figure 8 sensors-21-00957-f008:**
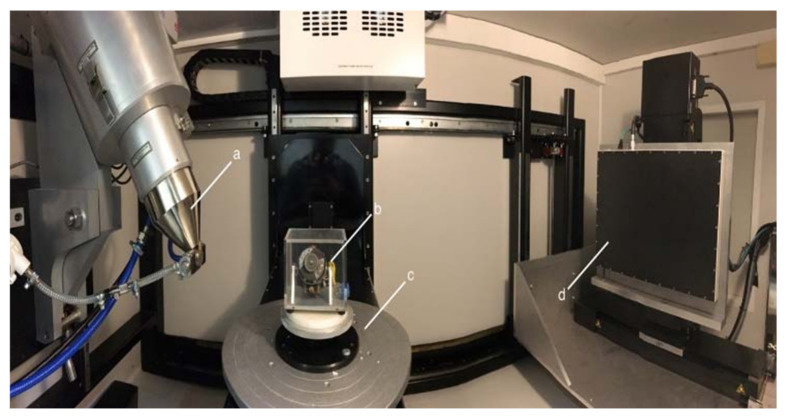
Measurement setup in CT: (**a**) X-Ray tube; (**b**) object to be scanned; (**c**) rotation table; and (**d**) flat panel detector.

**Figure 9 sensors-21-00957-f009:**
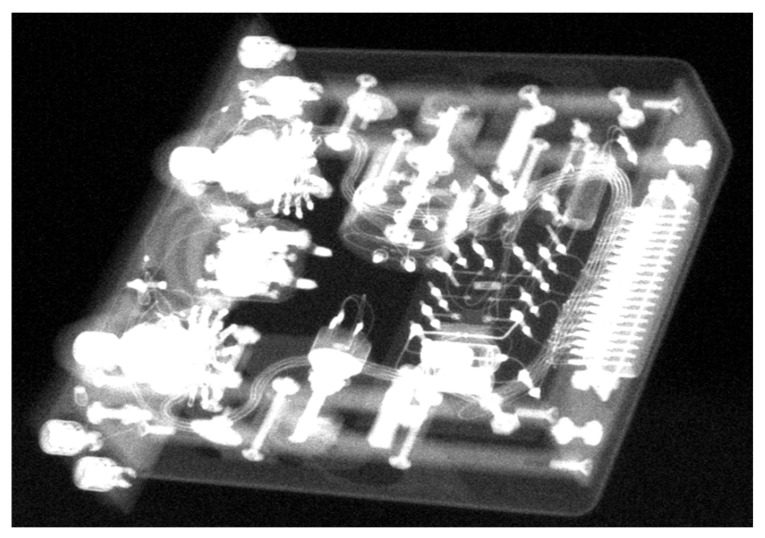
X-ray projection image with simulated and added Poisson noise.

**Figure 10 sensors-21-00957-f010:**
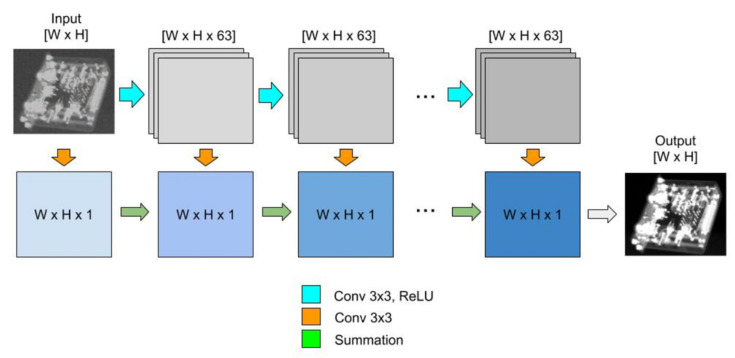
CNN architecture according to [32].

**Figure 11 sensors-21-00957-f011:**
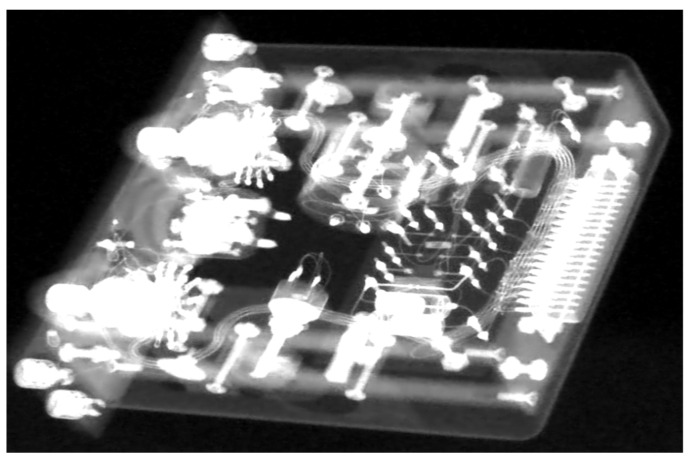
Result of the denoising of KH-09-09 using the CNN of Figure 10.

**Figure 12 sensors-21-00957-f012:**
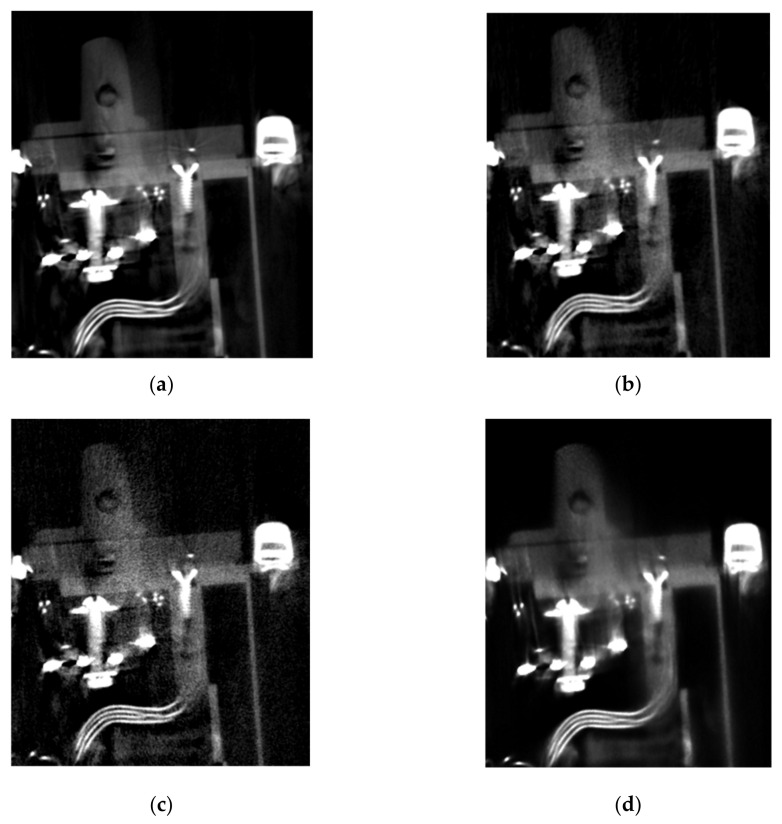
Region of volume KH-09-09 reconstructed: (**a**) FBP of high-dose projections; (**b**) FBP of denoised projection by CNN architecture of Figure 10; (**c**) FBP of noisy projections; and (**d**) MLEM of noisy projections.

**Figure 13 sensors-21-00957-f013:**
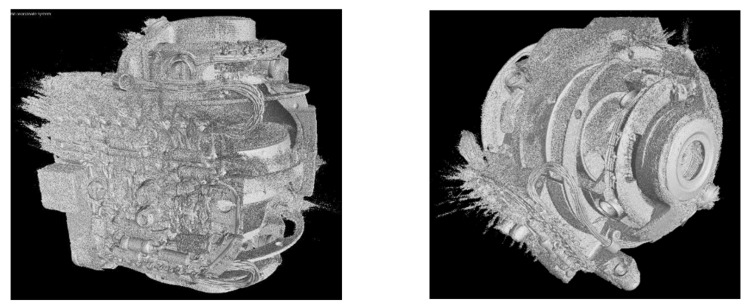
Two views of the iso-surface of G200 from the 3D volume data set reconstructed by the FBP.

**Figure 14 sensors-21-00957-f014:**
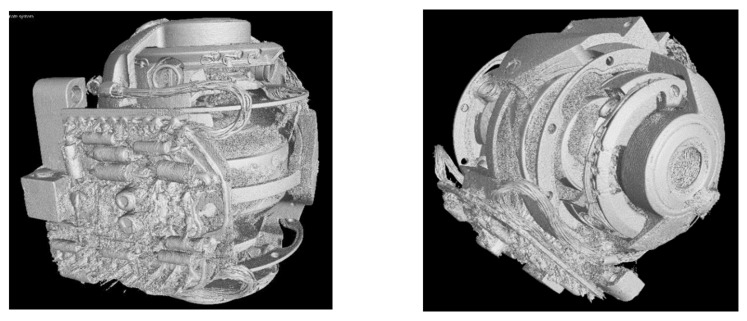
Two views of the iso-surface of the G200 calculated from the 3D volume data set reconstructed by the FBP from the denoised projections.

**Figure 15 sensors-21-00957-f015:**
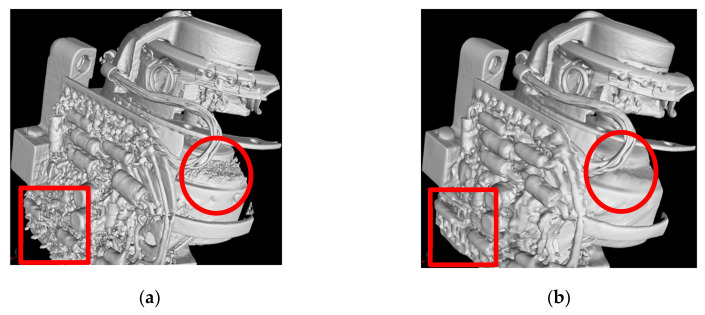
Comparison of two iso-surfaces calculated from the reconstructed volumes in the regions-of-interest (ROIs): (**a**) from original projections; and (**b**) from denoised projections.

**Figure 16 sensors-21-00957-f016:**
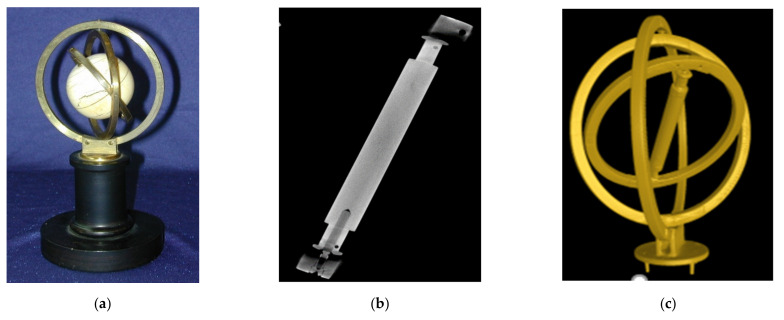
The Machine of Bohnenberger and CT scans: (**a**) photo; (**b**) cross-sectional view of the CT scan of the inner rod; and (**c**) gimbal iso-surface of the CT scan.

**Figure 17 sensors-21-00957-f017:**
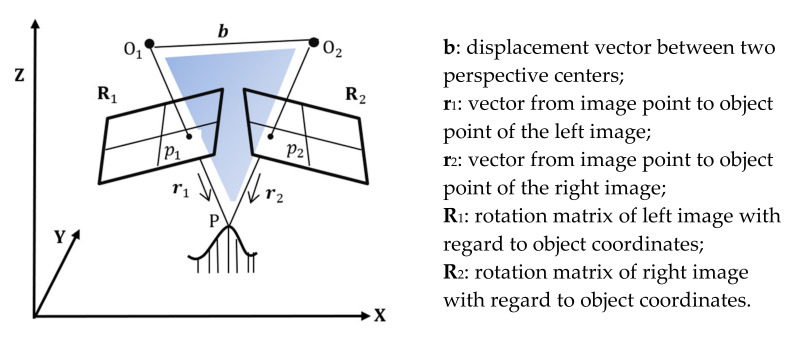
Basic principle of 3D point reconstruction using images: geometry of coplanarity.

**Figure 18 sensors-21-00957-f018:**
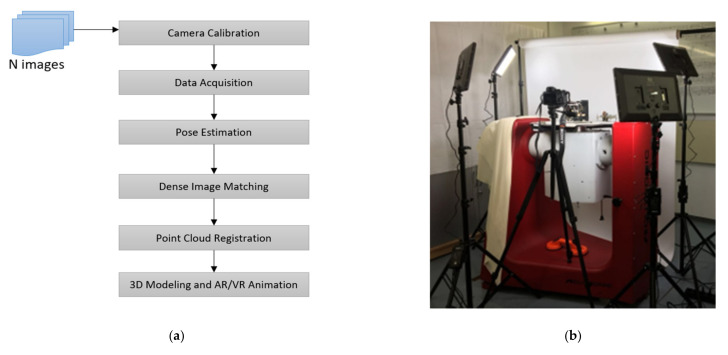
Data acquisition in the Gyrolog project: (**a**) workflow of CV/photogrammetry; and (**b**) Gyrolog Lab facility used for photo collections.

**Figure 19 sensors-21-00957-f019:**
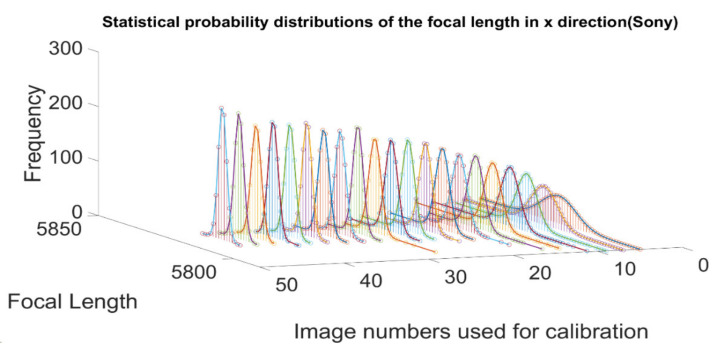
Gaussian fitting experiment results of the calibrated focal length in x direction for the Sony α7R II.

**Figure 20 sensors-21-00957-f020:**
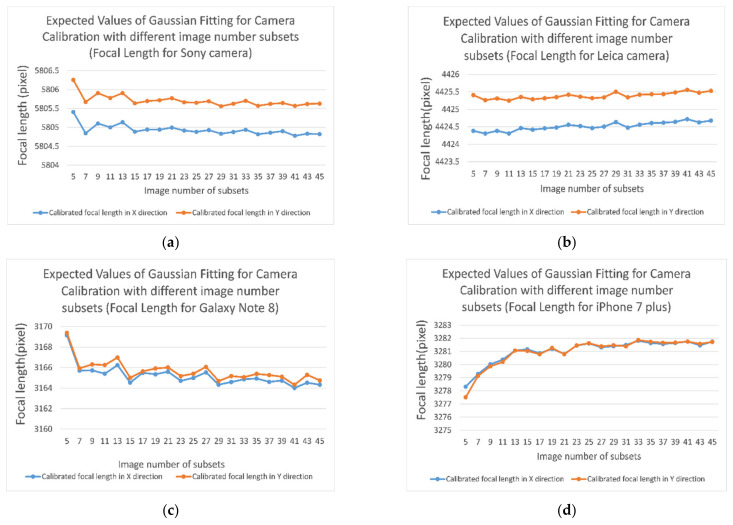
Gaussian fitting standard deviations for the calibrated focal length of four camera systems: (**a**) Sony α7R II focal length; (**b**) Leica Q focal length; (**c**) Samsung Galaxy Note 8 focal length; and (**d**) Apple iPhone 7Plus focal length.

**Figure 21 sensors-21-00957-f021:**
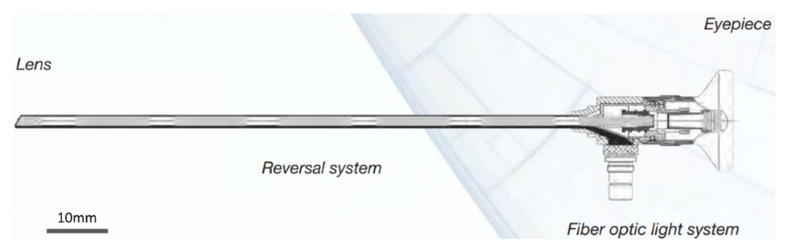
Structure of an endoscope. © Karl Storz SE & Co KG.

**Figure 22 sensors-21-00957-f022:**
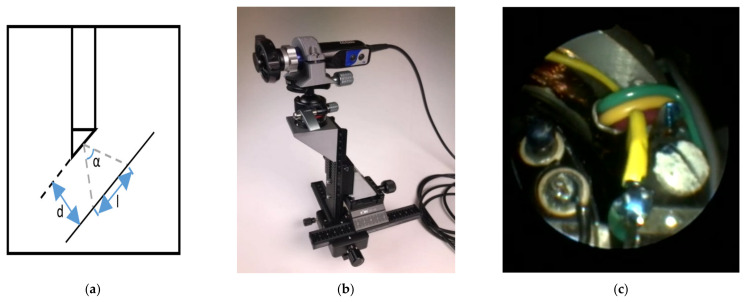
Endoscopic imagery: (**a**) opening angle; (**b**) mechanical gimbal set-up; and (**c**) image.

**Figure 23 sensors-21-00957-f023:**
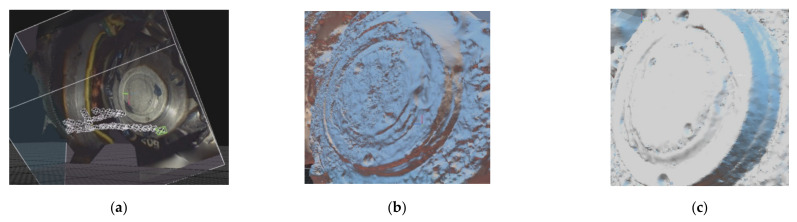
An endoscopy 3D mesh aligned with RealityCapture: (**a**) alignment; (**b**) meshed point cloud; and (**c**) meshed point cloud by CV/photogrammetry.

**Figure 24 sensors-21-00957-f024:**
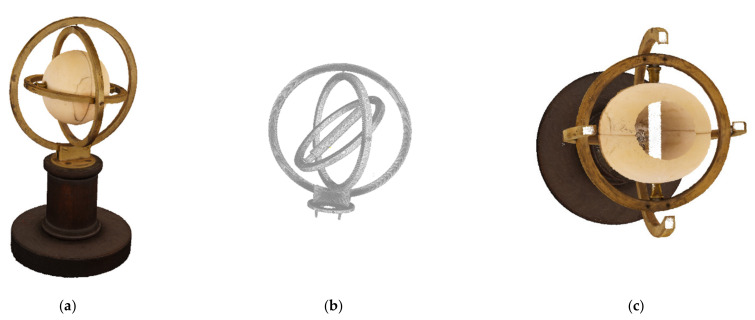
The 3D digital twin of the Machine of Bohnenberger: (**a**) meshed point cloud of CV/photogrammetry; (**b**) iso-surface of the gimbal CT scan; and (**c**) split view of the CT/CV integrated model.

**Figure 25 sensors-21-00957-f025:**
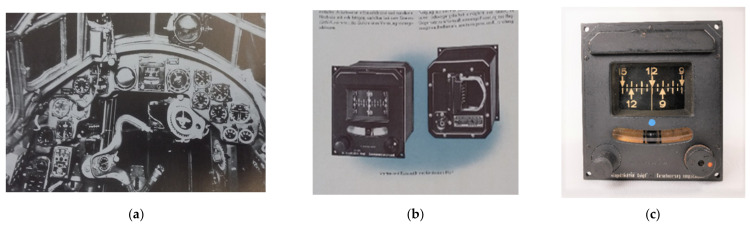
The LKu4 by Siemens: (**a**) mounted in a Junkers JU 88 A-1 cockpit; (**b**) a Siemens advert from the early 1940s; and (**c**) front view of the Gyrolog asset.

**Figure 26 sensors-21-00957-f026:**
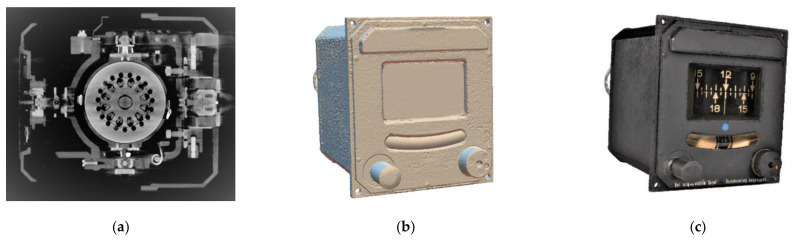
A 3D digital twin of the LKu4: (**a**) CT scan; (**b**) CV/photogrammetry point cloud mesh with normal and sprayed images; and (**c**) the CT/CV integrated model/.

**Figure 27 sensors-21-00957-f027:**
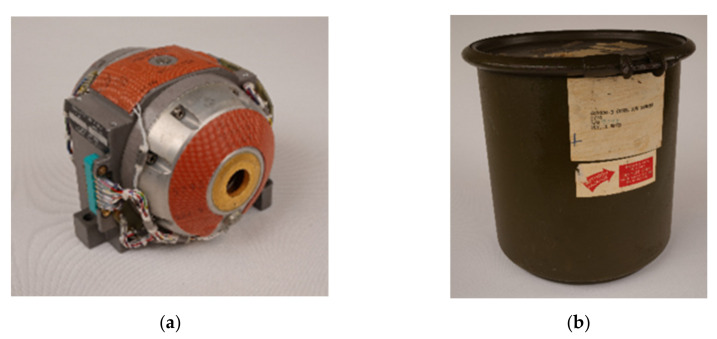
The Gyro G200 of the LN3 Inertial Platform: (**a**) Gyro G200; and (**b**) G200 transportation box.

**Figure 28 sensors-21-00957-f028:**
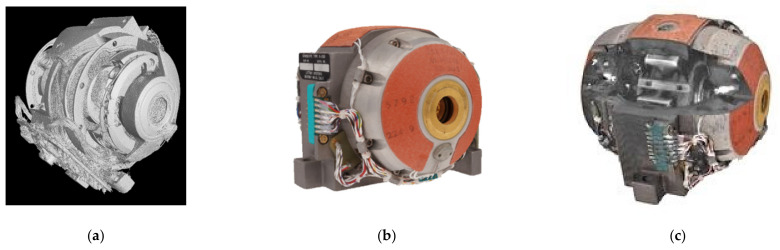
The 3D digital twin of the DM-LN3-G200 gyro: (**a**) iso-surface of a denoised FBP CT scan; (**b**) meshed point cloud of CV/photogrammetry; and (**c**) split view of the CT/CV integrated model.

**Figure 29 sensors-21-00957-f029:**
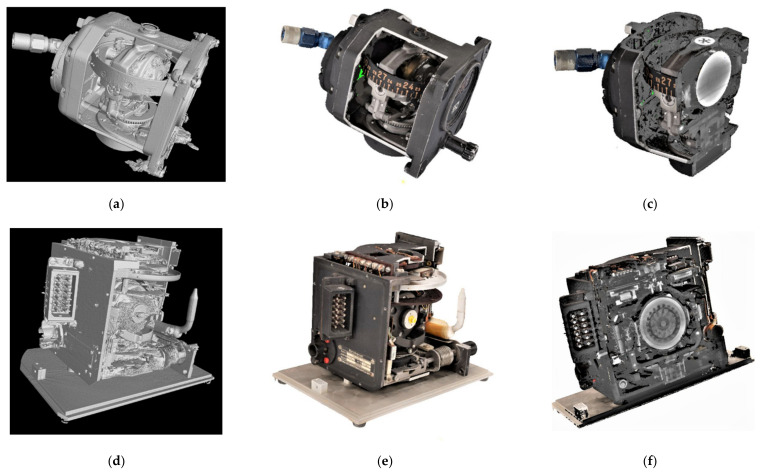
Two more 3D digital twins of the Stuttgart gyroscope collection—the Ternstedt direction gyro (**a**–**c**), and the Siemens direction gyro (**d**–**f**): (**a**,**d**) iso-surfaces of the denoised FBP CT scans; (**b**,**e**) meshed point clouds of CV/photogrammetry; and (**c**,**f**) split views of the CT/CV integrated models.

**Figure 30 sensors-21-00957-f030:**
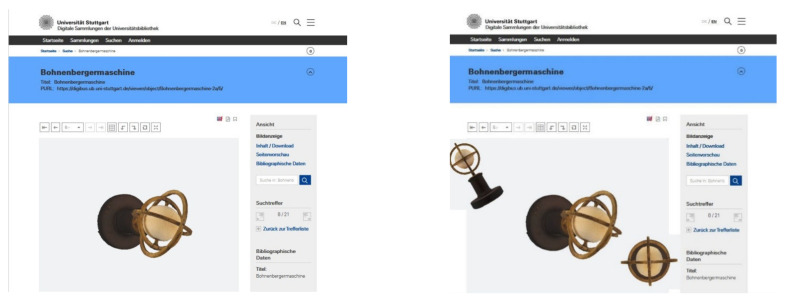
The Machine of Bohnenberger in two successive screen shots.

**Table 1 sensors-21-00957-t001:** CT scan parameters of the KH-09-09 specimen.

X-ray tube voltage (kV)	180
Current (µA)	400
Exposure (s)	1
Filter material (mm)	Copper (2.5)
Number of projections	1150

**Table 2 sensors-21-00957-t002:** CT scan parameters for the object G200.

X-ray tube voltage (kV)	170
Current (µA)	740
Exposure (s)	1.4
Filter material	Copper (4.5)
Number of projections	1256

**Table 3 sensors-21-00957-t003:** Tracing the gyroscope manufacturers [45].

Inventory Number	Manufacturer
KK02-09	Luftfahrtgerätewerk Hakenfelde GmbH, Berlin
KK09-09	Siemens App. u. Masch. GmbH, Berlin
KK12-09	Luftfahrtgerätewerk Hakenfelde, Berlin
KK34-20	hdc Luftfahrtgerätewerk Hakenfelde, Berlin

**Table 4 sensors-21-00957-t004:** Precision values and file sizes of some Gyrolog 3D digital twins.

Object	#CPs	σ_0_	σ_CT_	σ_CV_	CT Resolution	CV Resolution	File Size CV/Photogr.	Integrated File Size
G200	4	1.38	0.10 mm	0.08 mm	0.22 mm	0.05 mm	425 MB	1.05 GB
	10	1.07	0.08 mm	0.06 mm				
Machine of B.	4	1.36	0.12 mm	0.15 mm	0.035 mm	0.03 mm	157 MB	378 MB
	10	1.02	0.09 mm	0.11 mm				
Ternstedt Gyro	4	1.24	0.11 mm	0.14 mm	0.035 mm	0.03 mm	152 MB	1.46 GB
	10	1.09	0.10 mm	0.12 mm				

**Table 5 sensors-21-00957-t005:** Data volumes of different data formats.

Object Name	**.obj* File	**.gltf* File	**.gltf/Draco* File
G200	425 MB	154 MB	21.4 MB
Machine of B.	157 MB	59 MB	8.0 MB
Siemens Gyro	495 MB	120 MB	7.6 MB
Siemens LKu4	232 MB	91 MB	11.7 MB
Ternstedt Gyro	152 MB	66 MB	22.2 MB

## Data Availability

The data presented in this study are available via Goobi and DaRUS of the University of Stuttgart, Germany.

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
