# Peer review of "Making Historical Gyroscopes Alive—2D and 3D Preservations by Sensor Fusion and Open Data Access"

_sensors, 2021, doi:10.3390/s21030957_

Round 1

Reviewer 1 Report

This was a challenging read and the authors can be assured that I read the entire paper twice.

It is difficult for me to see the explicit aim of the paper. It read like an attempt to compress an entire research project (Gyrolog?) into a single paper.  If so, the authors do themselves a great disservice. A project should generate several papers, perhaps the authors could consider one or two aspects only for the moment.

I would like to see statements in the abstract that identify the impact, significance, rigor and novelty of the research.

There are numerous acronyms in the paper, and I think inclusion of a glossary would be helpful.

A major weakness, it seems to me, is a lack of structure. The authors may wish consider adapting the following.

  • Abstract
  • Introduction (including critical literature review)
  • Methodology
  • Results
  • Evaluation and discussion
  • Conclusions and recommendation for further research.

I would also encourage the authors to write in clear and succinct language, please put yourselves in the place of the readers.  I think this would commend your paper to the widest audience in order to get the knowledge out.

There is not a clear structured methodology in the paper.  What did the authors’ do? Why did they do it?  How did they do it? What did they find? What is the significance of the results?  What are its implications?

There are numerous formatting and other issues with the paper, I have marked some of them up and am happy to share my marked up document with the authors,

Author Response

See the extra response letter, thank you for your efforts!

Reviewer 2 Report

According to the authors, the work consists of the workflows for combining voxel representations and colored point clouds are described, to create Digital Twins of the tangible assets. Certainly, this reviewer is confused because the work can be considered at first a review or state of the art of the artifacts and later the authors make an attempt at the methodology that does not quantify nor does a discussion of the results exist. There's a big hype on the topic, so the paper should be rock-solid to be well accepted by the scientific community. Below you'll find various comments, referring to some lines that need some rework. I do believe that most of the issues are related to an over-simplification while writing the paper, but please try to make them more consistent.

In page 2, line 69, the first time the photogrammetry is located, use the abbreviations and what happens with the Multi-View-Stereo

In page 2, line 72-74 is confusing, please clarify the terms.

In page 3, it is convenient on line 93 to cite software like C2C.

In page 3, in line 97 you must cite important works.

In page 13, in the basis that this premise of line 398-399 is established

Author Response

see the Response Letter, thank you for your efforts.

Reviewer 3 Report

The proposed paper illustrates some procedures used to create digital twins of some historical instruments, through the combined use of technologies including CT, CV and photogrammetry, with the aim of making the results publicly shared through Open Data platforms.

The approach described is particularly interesting and well structured in the presentation, starting from the introduction to the large number of technologies covered and correctly referenced. The challenges faced for the digitization of the studied objects are well described both in the methodological approach and with abundant examples.

I have only a few suggestions for improving the quality of the article.

  • Minor English language spell check is required, such as the use of “und” at line 29 in the Abstract.
  • Some Figures [17, 19] do not have a good resolution and are therefore not very readable, but I assume that in the final version this problem will be corrected
  • Introducing the glTF files [line 705] it would be advisable to specify them in some more detail

Author Response

See the Response Letter, thank you for your efforts.

Round 2

Reviewer 1 Report

Many thanks to the authors for their resubmission.  I have re-read the paper and the authors' comments on the first review.

This is a much stronger paper, clear (at least to me) and now easy to follow with a more robust structure and methodology.  I am satisfied that they have addressed my earlier comments.

A few minor things.  There is a reference in line 464 to Figure 18a but I cannot find Figure 18a in the paper, should this be Figure 18?

Line 528, should 1 be I?

Line 761 g1TF and compare line 766 g1tf.  Just a suggestion that the authors' nomenclature should be consistent.  I appreciate this is window dressing so to speak.  Perhaps another read through would help clear up these minor details.

Without wishing to insult the authors' intelligence, don't forget to remove the highlights and I do not require another submission before publishing.

Best wishes

Author Response

Please see the response letter attached!

Reviewer 2 Report

The authors have considerably improved the paper which makes it more robust. Even so, they should include a graphical scale in figure 21 and review the magnitudes in table 4. I think that citation 34 should be improved. The CV / photogrammetry workflow could be cited and commented on in the scientific literature.

Author Response

(The authors gave the same response as above.)
